# Immigrant–Native Differences in Sugar-Sweetened Beverage and Snack Consumption and Preventive Behaviors Associated with Severe Early Childhood Caries: A Large-Scale Survey in Taiwan

**DOI:** 10.3390/ijerph16061047

**Published:** 2019-03-22

**Authors:** Ying-Chun Lin, Chin-Shun Chang, Pei-Shan Ho, Chien-Hung Lee, Jen-Hao Chen, Hsiao-Ling Huang

**Affiliations:** 1Department of Oral Hygiene, College of Dental Medicine, Kaohsiung Medical University, Kaohsiung 807, Taiwan; bonnie0925.tw@gmail.com (Y.-C.L.); psho@kmu.edu.tw (P.-S.H.); 2Taiwan Society of Oral Health, Keelung 201, Taiwan; deway@ms25.hinet.net; 3School of Oral Hygiene, College of Oral Medicine, Taipei Medical University, Taipei 110, Taiwan; 4Division of Medical Statistics and Bioinformatics, Department of Medical Research, Kaohsiung Medical University Hospital, Kaohsiung 807, Taiwan; cnhung@kmu.edu.tw; 5Department of Public Health and Research Center for Environmental Medicine, Kaohsiung Medical University, Kaohsiung 807, Taiwan; 6School of Dentistry, College of Dental Medicine, Kaohsiung Medical University, Kaohsiung 807, Taiwan; jehach@kmu.edu.tw; 7Department of Prosthodontics, Kaohsiung Medical University Hospital, Kaohsiung 807, Taiwan

**Keywords:** severe early childhood caries (SECC), caries prevention, tooth brushing, flossing, fluoride varnish, sugar-sweetened beverages (SSBs), snacks, immigrant

## Abstract

This study assessed the differences between immigrants and natives in terms of combined effects of sugar-sweetened beverage (SSB) or snack consumption and preventive behaviors for severe early childhood caries (SECC) through a large-scale survey of 31,565 native and 1046 immigrant child–parent pairs in Taiwan. Children aged 3–5 years underwent dental examinations, and parents completed structured questionnaires. Immigrants had a significantly higher SECC prevalence than native children (32.3% vs. 19.4%). A combined effect of SECC was observed in native children who did not receive assistance when brushing teeth at night before sleep and those who consumed SSBs more than four times weekly (adjusted odds ratio (aOR) = 4.8). Moreover, native children who did not use dental floss and who consumed snacks more than four times weekly had an aOR of 4.1 for SECC. The combined effects of children with immigrant parents who did not receive assistance when brushing their teeth at night before sleep and those who consumed snacks more than four times weekly had an aOR of 8.2 for SECC. The results suggest the necessity of cross-cultural caries prevention programs for immigrants. Parents must limit children’s SSB and snack intake, and implement preventive measures to reduce SECC development.

## 1. Introduction

Dental caries in young children is a public health concern. The premature loss of teeth in the primary dentition of children may cause malocclusion [1]. Tooth loss has negative effects on masticatory ability and nutrient intake, and it increases the risk of malnutrition. Drury et al. [2] described the diagnostic criteria for severe early childhood caries (SECC). Children with SECC are at risk of anemia and iron deficiency [3]. Moreover, their physical development and quality of life are affected [4]. The Taiwan National Oral Health Survey in 2011 estimated SECC prevalence to be 27.5%, 49.0%, 58.7%, and 58.3% in children younger than three, and those aged three, four, and five years, respectively [5].

Women from southeast Asian countries have been migrating to Taiwan since 1987. The majority of immigrant women are Vietnamese and Indonesian, followed by Thais, Filipinos, and Cambodians. These women are colloquially called “foreign brides” or “alien brides”, because their marriages are arranged by marriage brokers. Since 2010, an average of 3.5% of children per year in Taiwan are born to a foreign spouse [6]. This particular group of immigrant women is highly susceptible and vulnerable to health problems because of language barriers, cultural conflicts, social and interpersonal isolation, and a lack of support systems. This segregation gradually resulted in inferior medical care for these women and their children [7]. Recent studies reported that immigrant mothers have less caries-related knowledge, a negative attitude toward oral hygiene, and less frequent positive oral health behavior [8], and the primary dentition caries experience is significantly higher among immigrant children, indicating disparities between the oral health of immigrants and native children [9].

A higher consumption frequency of sugar-sweetened beverages (SSBs) or snacks is associated with adolescent metabolic dysfunctions and a higher level of deciduous caries [10,11,12,13,14]. Young children who consumed snacks three and more times daily exhibited more carious lesions (80%). Furthermore, 61.7% of children affected by caries preferred sticky food [13]. Young children who consumed three or more SSBs daily had, on average, 47.1% more decayed, missing, and filled deciduous teeth than those who did not consume sweetened beverages [14]. The brushing of teeth, flossing, regular dental visits, and fluoride varnish application are evidenced as oral health behaviors that prevent dental caries [15]. Most parents in Taiwan exhibit inappropriate oral health behaviors. For example, a study found that children are generally rewarded with candies, cookies, and sweetened beverages, and one in four parents does not advise their children to brush their teeth after eating sugary foods [16]. This behavior may be associated with caries development in young children.

Taipei is the capital city of Taiwan. In comparison to other areas, Taipei city has one of the highest densities of chain convenience stores [17] and hand-shaken beverage shops, thus providing the population with easy access to snacks and SSBs. They also have the highest density of dental clinics, thus providing accessibility to and availability of dental medical resources. The National Health Insurance provides free fluoride varnish every six months for children younger than six years, and Taipei is the target city of health policy implementation. Therefore, the caries prevalence of primary dentition is lower in Taipei city than in other areas [5]. Immigrant women have the largest population in New Taipei City, followed by Taipei City. In this study, a large representative group of native children and children born to a foreign spouse were surveyed. The prevalence of SECC, the frequency of SSB and snack consumption, and the caries preventive behaviors were examined. This study aimed to investigate immigrant–native differences in relation to SSB and snack intake and dental caries preventive behaviors, associated with SECC in Taipei, to develop dental caries prevention and intervention strategies for young children.

## 2. Materials and Methods

### 2.1. Study Design and Participants

This study integrated the findings of a large-scale survey of oral health conditions in Taipei, Taiwan. These young children were recruited from the 52 community health centers and from 659 preschools (149 public, 510 private) in 12 districts of Taipei, using the preschooler integrated community screening test, conducted by the Department of Health, Taipei City, Government of Taiwan, in 2015. There were 45,872 children aged 3–5 years who participated in this survey study. The children received dental examinations, and their parents completed a structured self-administered questionnaire. Invalid questionnaires were excluded, for example, when parents did not fill in the demographic items (e.g., maternal ethnicity and parents’ educational level). The final datasets, obtained from 32,611 (31,565 native children and 1046 children born to a foreign spouse) child–parent pairs, matched completely the questionnaire and oral health data. Written informed consent was obtained from the parents, before examining the children, and the privacy rights of the participants were strictly complied with. This study was approved by the Institutional Review Board of Taipei City Hospital (TCH-IRB-1030114).

### 2.2. Instrument

The survey instrument was adopted and modified using an established validating questionnaire from the published literature [18]. The participating parents completed a self-administered questionnaire, including closed-ended items with dichotomous, ordinal, and multiple-choice responses, to assess relevant demographic, SSB and snack intake, and caries preventive behavior variables. Standardized dental examinations ([19] pp. 42–55) were conducted by trained and calibrated dentists. Caries and tooth fillings were visually detected using an operating light, a dental mirror, and the World Health Organization community periodontal index screening probe to determine whether observed small dark pits were caries or stains. Finally, a dentition status and a treatment requirement checklist were used to record the oral health status of the children.

### 2.3. Dependent Variable

Teeth that were decaying, temporarily filled, or damaged by caries were indicated as “d”; teeth extracted because of decay or because they could not be filled were indicated as “m”; and permanently filled teeth were indicated as “f”. The combined data comprised the decayed, missing, or filled teeth (dmft) score of each child. In the study, children aged 3–5 years with one or more decayed or missing (because of caries) tooth, or with a filled smooth surface in the primary maxillary anterior teeth, as well as those with a dmft score of 4, 5, or 6 (for ages of three, four, and five years, respectively) or more, were diagnosed as having SECC [2]. Accordingly, the participating children were categorized into caries-free, non-SECC, and SECC groups.

### 2.4. Independent Variables

#### 2.4.1. SSB and Snack Consumption

Five items measuring children’s consumption frequencies of SSBs and snacks were taken from the previous literature [18]. Children’s SSB and snack consumption was measured based on their parents’ reports. The number of times in the previous week that the children consumed the following items was reported: (1) soft drinks (e.g., soda); (2) milk yogurt (e.g., probiotic drinks); (3) hand-shaken drinks (e.g., bubble milk tea); (4) sweet snacks (e.g., cake, candies); and (5) salty snacks (e.g., potato chips). The question was worded as follows: “How many times did your child consume_______in the previous week?” (with possible answers being “less than once”, “one to three times”, “four to six times”, and “seven times or more”). The average SSB or snack consumption of a child in a typical week was categorized as follows: less than once, one to three times, and more than four times.

#### 2.4.2. Caries Preventive Behaviors

Five items measuring the caries preventive behaviors of parents and their children were employed. The operational definition of the caries preventive behaviors of parents and their children was established using the following questions: “Does your child need to be reminded to brush their teeth? (Yes/No);” “Do you assist your child in brushing his/her teeth before sleep at night? (Yes/No);” “Do you assist your child in cleaning his/her teeth by using dental floss? (Yes/No);” “Do you take your child for semiannual dental visits? (Yes/No);” and “Does your child receive fluoride varnish application every six months? (Yes/No).”

### 2.5. Data Collection

The dental examinations were arranged by the Department of Health, Taipei City. The preschool teachers gave parents the consent forms and questionnaires. If parents agreed, their children participated in this survey, and the parents had to answer all questions completely and bring the questionnaire to the preschool. Then, the child received a dental examination by the dentist at the preschool. If the child did not go to a preschool, the public nurse informed parents to take their children to the community health centre at a specific time. The dentists checked the children’s oral health status, while the parents completed the questionnaire.

### 2.6. Statistical Analysis

The outcome of children’s dental caries was categorized as caries-free, non-SECC, and SECC. Means, standard deviations, and percentages were used to express the distribution of study variables, and an ANOVA and χ^2^ test were employed to assess the relationship between continuous and categorical factors and dental caries conditions. Because the outcome was trinary, multinomial logistical regression models were used to determine the risk of developing non-SECC or SECC in relation to caries-free participants. The adjusted odds ratio (aOR), with 95% confidence intervals (CIs), which was obtained by the exponentiation of the corresponding regression coefficient, was used to evaluate the association between the study variables and a particular dental caries status, after adjusting for the effects of potential confounding variables (child’s age and gender, and parent’s educational level). Multiplicative logistical regression models were used to evaluate the main and combined effects of SSB and snack intake, as well as the dental caries preventive behaviors, on children’s dental caries outcome.

## 3. Results

Table 1 presents the primary tooth status of native and immigrant children aged 3–5 years, who participated in this study. In total, 19.8% of children were diagnosed with SECC, with a higher prevalence in immigrant children than in native children (32.3% vs. 19.4%). The means of decayed teeth (5.38) and dmft (8.47) were higher for immigrant children than for the native children (4.37 and 8.10, *p* < 0.001 and *p* = 0.01, respectively). However, the native children had a higher mean of filled teeth than the immigrant children (3.50 vs. 2.79, *p* < 0.001). Table 2 shows the demographic backgrounds of the children aged 3–5 years and associated dental caries conditions. The age of the children and level of parental education were associated with the status of dental caries in native and immigrant participants, with higher proportions of non-SECC and SECC in children aged five years and lower proportions of SECC in children whose parents had a postgraduate or higher educational level. Gender was also related to the caries condition in native participants; boys exhibited a higher percentage of SECC than girls (54.4% vs. 45.7%). The immigrant group had a lower parental educational level than the native group among children who were classed as caries-free, non-SECC, and SECC (*p* < 0.001).

Table 3 presents the association between SSB and snack consumptions and dental caries conditions in the children. Adjusted for covariates, a higher intake of soft drinks, hand-shaken drinks, and sweet snacks was significantly associated with a higher risk of developing SECC in native children. Compared with native participants in the non-SECC group, the children who consumed soft drinks, hand-shaken drinks, and sweet snacks more than four times per week had a 1.7-, 1.8-, and 2.0-fold higher risk of developing SECC. In immigrant participants, children who consumed soft drinks one to three times per week had a 1.7-fold higher risk of non-SECC than children who consumed soft drinks less than once per week. Compared with immigrant participants with non-SECC, children who consumed milk yogurt one to three times and more than three times per week had a 1.5- and 1.9-fold higher risk of developing SECC, respectively. Children with immigrant parents, compared to children with native parents, had a 1.4- and 2.1-fold higher likelihood of developing SECC when consuming milk yogurt one to three times and more than three times per week, respectively.

Table 4 presents the relationship between dental caries preventive behaviors and the caries status of the children. In native participants, children who needed to be reminded to brush their teeth (aOR of 1.2 and 1.4), required parents’ assistance in teeth brushing at night or before sleep (aOR of 1.1 and 1.2), had regular semiannual dental visits (aOR of 1.2 and 1.5), and did not receive semiannual fluoride varnish applications (aOR of 1.1 and 1.3) had a higher risk of developing non-SECC and SECC than the children with caries-free teeth. Moreover, a higher risk ratio of having SECC than contracting non-SECC was also observed in children who needed to be reminded to brush their teeth, required parental assistance in brushing teeth at night or before sleep, did not use dental floss, had semiannual dental visits, and did not receive fluoride varnish every six months (aOR ratios of 1.4, 1.2, 1.1, 1.5, and 1.3, respectively). Among the immigrant participants, the children who were not assisted by their parents in brushing teeth before sleep and who had semiannual dental visits exhibited a 1.6- and 1.7-fold higher likelihood of developing SECC, respectively. Children of immigrant participants who did not use dental floss had a significantly higher likelihood of developing SECC compared with the native participants (aOR of 1.6).

The combined effects of dental caries preventive behaviors and the SSB and snack intake of the native children with SECC were analyzed (Table 5). The significant combined effects of the native children, who did not receive parental assistance in brushing teeth at night or before sleep, and who consumed SSBs more than four times per week, yielded an aOR of 4.8 (95% CIs: 3.3–7.1) for SECC. Furthermore, the children who consumed snacks more than four times per week and who did not use dental floss yielded an aOR of 4.1 (95% CIs: 3.2–5.1) for SECC. Table 6 presents the significant combined effects for the immigrant children who did not receive parental assistance in brushing teeth at night or before sleep, and who consumed snacks more than four times per week, yielding an aOR of 8.2 (95% CIs: 1.7–38.8) for SECC. There was no interaction effect of caries preventive behaviors and SSB or snack intake associated with SECC among the two groups.

## 4. Discussion

The findings of this study validated oral health disparities between immigrant and native young children. Furthermore, the adverse correlation between SSB or snack consumption and SECC was studied. Significant combined effects were observed in the children who consumed SSBs or snacks four or more times per week, did not receive parental assistance when bruising teeth at night or before sleep, cleaned teeth without using dental floss, and did not have semiannual dental fluoride varnish application.

The results indicate a racial difference related to the dental caries status of young children. A significantly higher number of children with immigrant parents had untreated caries teeth, compared with native children, and almost one out of three had SECC. Nevertheless, the average number of filled teeth was low in immigrants, which demonstrated that the immigrant children had worse oral health and higher dental treatment requirements [9]. Children from minority families have worse oral health outcomes and use oral healthcare services less frequently. A previous study reported [20] that oral health literacy in mothers from families with a low socioeconomic status was low. While the National Health Insurance actively promotes free fluoride varnish every six months for children aged less than six years, health information is not readily available for people in low socioeconomic segments, particularly for immigrant mothers with a language barrier.

This study indicated that the children who consumed soft drinks, hand-shaken drinks, milk yogurt, and sweet snacks more than four times per week exhibited a higher risk of developing SECC. Taiwan has one of the highest densities of chain convenience stores worldwide, with an average density of one store per 0.26 km^2^ in Taipei. People who live in Taipei are, therefore, easily able to obtain SSBs or snacks [17]. Among Taiwanese children, almost 90% of five-year-old children consume SSBs and snacks once a day, and sugar consumption increases with age [21]. A recent study reported that high sugary foods have an addictive potential, and the parental influence on their children’s diet is high [22]. The children whose mothers were immigrants exhibit a higher frequency of SSBs consumption than the children with native mothers. For children with SECC who were born to immigrant mothers, 80% of them consumed milk yogurt (e.g., probiotic drinks) more than one time per week. Probiotic drinks are the cheapest drink compared with soft drinks and hand-shaken drinks. A previous study indicated that women with low income and education are particularly vulnerable to diets with high added sugars [23]. When children begin to make their own decisions about sugary food consumption, a low education level in mothers contributed to a lack of information on early childhood caries prevention, and these risk factors cause children’s dental caries [24].

In the present study, we found that over half of children with SECC required their parents to remind them to brush their teeth. Less than half of the parents of these children helped their children brush their teeth at night or before sleep. Murthy et al. [25] reported that, for children under six years of age, tooth-brushing should be performed by parents. Behaviors for preventing SECC include parental assistance in tooth-brushing at night and the use of dental floss. Tooth-brushing could reduce carious lesions on smooth surfaces, and flossing in toddlers is valuable for the prevention of interdental caries. Currently, the prevalence of interdental caries in young Taiwanese children is high. However, less than half of parents in Taiwan use dental floss to clean their children’s interdental surfaces [8,9,16]. Similarly, in this study, regardless of the group, just 22% of immigrant parents and 41% of native parents used dental floss to clean their children’s interdental surfaces. The caries-inhibiting effect of fluoride varnish application is evident, and semiannual fluoride varnish applications could reduce cases of tooth decay in primary dentition [26]. Since 2012, dentists provide free fluoride varnish applications every six months to children aged <6 years in kindergartens. In this study group, 50.3% of immigrant children and 62.7% of native children received fluoride varnish application. We observed that, regardless of whether those with SECC were native or immigrant children, the percentage who received fluoride varnish applications was higher than the percentage of parents who assisted in tooth-brushing at night or before sleep, or who flossed for their children. That means that there is a high prevalence of young children receiving semiannual fluoride varnish applications in kindergarten. However, parents of children with SECC exhibited lower daily caries preventive behaviors toward their children, especially immigrant parents.

The combined effects revealed that native children with SECC consumed SSBs four or more times per week and did not receive parental assistance in tooth-brushing at night or before sleep, and children who consumed snacks four or more times per week generally did not floss. Children whose mothers were immigrants, those whose parents did not assist them with teeth brushing at night or before sleep, and those who consumed snacks four or more times per week had an 8.2-fold increase in the likelihood of developing SECC. Public Health England ([15], pp. 6–7) provided the following suggestions for the prevention of caries in children aged 0–6 years: (1) parents should brush children’s teeth or supervise tooth-brushing; (2) as soon as teeth erupt in the mouth, they should be brushed twice daily, immediately before bed and on one other occasion, with a fluoridated toothpaste; and (3) the frequency and quantity of sugary food and drinks should be minimized. The current study also indicates that children who consume SSBs and snacks frequently and who are not given parental assistance in tooth-brushing and flossing might have an increased risk of developing SECC.

The present study had some limitations. Firstly, SSB and snack consumption data were obtained from a self-administered questionnaire, completed by parents. The participating children remained in preschools during the daytime five days a week. Therefore, the parents may have underreported the SSB and snack consumption of their children. However, the increased proportion of weekly SSB and snack consumption in the diets of children does not change the nature of the association. By contrast, it emphasizes the oral health implications of the present findings. Secondly, the dentists did not examine the intra- and inter-reliability before performing the dental examination on the children; however, they were well-trained in pediatric dentistry. Therefore, the dental caries diagnosis was reliable. Thirdly, only 71% of child–parent data were analyzed, because some preschool teachers did not check the integrity of the questionnaire or children’s absences for the dental examination at the preschool. However, the 3.2% of children born to a foreign spouse in this study is similar to the ratio in the national young children’s population. Thus, this large-scale survey is sufficient to represent the young children’s population in Taiwan. Finally, because of the cross-sectional nature of this study, causal inferences were not drawn.

## 5. Conclusions

Differences in immigrant and native children, in terms of the oral health status, SSB consumption, and dental caries preventive behaviors of parents toward their children, were observed. The adverse combined effects of SSB and snack consumption on SECC prevalence varied based on the preventive behaviors of the children. The findings suggest an implementation of cross-cultural competence caries prevention programs for these immigrant minorities to increase oral health literacy, reducing disparities in the child oral healthcare of immigrant and native children. Furthermore, limiting the consumption frequency of SSBs and snacks among young children, emphasizing parental assistance when brushing teeth before sleep, cleaning teeth using dental floss, and receiving semiannual protective fluoride varnish application must be publicized to improve the knowledge of the parents of young children, thus further reducing SECC development.

## Figures and Tables

**Table 1 ijerph-16-01047-t001:** Primary tooth status of children aged 3–5 years in relation to dental caries status in native and immigrant participants.

	Native				Immigrant				Total				
Factor	Caries-Free	Non-SECC ^1^	SECC ^1^	Diff. ^2^	Caries-Free	Non-SECC ^1^	SECC ^1^	Diff. ^2^	Caries-Free	Non-SECC ^1^	SECC ^1^	Diff. ^2^	*p*-Value ^3^
**Participants**	13,666 (43.3%)	11,767 (37.3%)	6132 (19.4%)		334 (31.9%)	374 (35.8%)	338 (32.3%)		14,000 (42.9%)	12,141 (37.3%)	6470 (19.8%)		<0.001
**Primary Tooth Status, mean ± SE**													
decayed teeth	0	1.45 ± 0.01	4.37 ± 0.05	2.92 *	0	1.59 ± 0.07	5.38 ± 0.22	3.79 *	0	1.45 ± 0.01	4.42 ± 0.04	2.97 *	<0.001
missing teeth	0	0.11 ± 0.00	0.23 ± 0.01	0.12 *	0	0.13 ± 0.03	0.30 ± 0.05	0.17 *	0	0.11 ± 0.00	0.23 ± 0.01	0.12 *	0.045
filled teeth	0	0.83 ± 0.01	3.50 ± 0.04	2.67 *	0	0.78 ± 0.06	2.79 ± 0.17	2.01 *	0	0.83 ± 0.01	3.46 ± 0.04	2.63 *	<0.001
dmft	0	2.39 ± 0.01	8.10 ± 0.04	5.71 *	0	2.50 ± 0.06	8.47 ± 0.18	5.97 *	0	2.39 ± 0.01	8.12 ± 0.04	5.73 *	0.010

**Abbreviations**: SECC: severe early childhood caries; SE: standard error; dmft: total number of teeth (t) that are decayed (d), missing (m), and filled (f); * *p* < 0.05. ^1^ Non-SECC group is defined as 0 < dmft < 4 for an age of 3 years, 0 < dmft < 5 for an age of 4 years, and 0 < dmft < 6 for an age of 5 years. SECC group is defined as dmft ≥4 for an age of 3 years, ≥5 for an age of 4 years, and ≥6 for an age of 5 years. ^2^ Diff. denotes the difference in means of decayed teeth, missing teeth, filled teeth, and dmft between SECC and non-SECC groups. ^3^
*p*-Values for percentages or mean differences between native and immigrant SECC groups.

**Table 2 ijerph-16-01047-t002:** Demographic backgrounds of children aged 3–5 years in relation to dental caries status in native and immigrant participants.

Factor	Native							Immigrant									
	Caries-Free		Non-SECC ^1^		SECC ^1^		*p*-Value	Caries-Free		Non-SECC ^1^		SECC ^1^		*p*-Value	*p*-Value ^2^	*p*-Value ^3^	*p*-Value ^4^
	No.	(%)	No.	(%)	No.	(%)		No.	(%)	No.	(%)	No.	(%)				
**Age**							<0.001							<0.001	0.035	0.039	0.070
3 years old	3950	(28.9)	1828	(15.5)	1103	(18.0)		78	(23.4)	42	(11.2)	53	(15.7)				
4 years old	5026	(36.8)	4263	(36.2)	2194	(35.8)		143	(42.8)	132	(35.3)	107	(31.7)				
5 years old	4690	(34.3)	5676	(48.2)	2835	(46.2)		113	(33.8)	200	(53.5)	178	(52.7)				
**Gender**							0.012							0.849	0.330	0.428	0.378
Female	6547	(47.9)	5530	(47.0)	2799	(45.7)		151	(45.2)	168	(44.9)	146	(43.2)				
Male	7119	(52.1)	6237	(53.0)	3333	(54.4)		183	(54.8)	206	(55.1)	192	(56.8)				
**Educational attainment**																	
**Mother’s educational level**							<0.001							<0.001	<0.001	<0.001	<0.001
High school or lower	1721	(12.6)	2061	(17.5)	1519	(24.8)		147	(44.0)	229	(61.2)	242	(71.6)				
College	8970	(65.6)	7570	(64.3)	3747	(61.1)		155	(46.4)	125	(33.4)	87	(25.7)				
Master/Doctor	2975	(21.8)	2136	(18.2)	866	(14.1)		32	(9.6)	20	(5.4)	9	(2.7)				
**Father’s educational level**							<0.001							<0.001	<0.001	<0.001	<0.001
High school or lower	1764	(13.3)	1979	(17.5)	1506	(25.6)		107	(33.1)	180	(50.1)	187	(57.7)				
College	7040	(53.1)	5985	(52.9)	2953	(50.1)		149	(46.1)	135	(37.6)	110	(34.0)				
Master/Doctor	4445	(33.6)	3355	(29.6)	1431	(24.3)		67	(20.7)	44	(12.3)	27	(8.3)				

Abbreviations: SECC: severe early childhood caries. ^1^ Non-SECC group is defined as 0 < dmft < 4 for an age of 3 years, 0 < dmft < 5 for an age of 4 years, and 0 < dmft < 6 for an age of 5 years. SECC group is defined as dmft ≥4 for an age of 3 years, ≥5 for an age of 4 years, and ≥6 for an age of 5 years. ^2^
*p*-Value for comparing demographic variables between the native and immigrant groups among caries-free children. ^3^
*p*-Value for comparing demographic variables between the native and immigrant groups among non-SECC children. ^4^
*p*-Value for comparing demographic variables between the native and immigrant groups among SECC children.

**Table 3 ijerph-16-01047-t003:** Adjusted odds ratios of dental caries status associated with sugar-sweetened beverages and snacks intake in native and immigrant participants.

Factor	Native						Immigrant						
	Caries-Free	Non-SECC		SECC		SECC vs. Non-SECC	Caries-Free	Non-SECC		SECC		SECC vs. Non-SECC	
	No.	No.	aOR ^1^ (95% CIs)	No.	aOR ^1^ (95% CIs)	aOR Ratio ^1^ (95% CIs)	No.	No.	aOR ^1^ (95% CIs)	No.	aOR ^1^ (95% CIs)	aOR Ratio ^1^ (95% CIs)	aOR^1,2^ (95% CIs)
**Sugar-sweetened beverages (SSBs)**													
**Soft drinks (e.g., soda)**													
less than once per week	11,186	8896	1.0	4240	1.0	1.0	258	241	1.0	210	1.0	1.0	1.0
1–3 times per week	2384	2743	1.2 (1.1–1.3)	1773	1.3 (1.2–1.4)	1.3 (1.2–1.4)	72	125	1.7 (1.1–2.6)	121	1.5 (0.9–2.2)	0.9 (0.6–1.2)	1.2 (0.9–1.6)
4+ times per week	96	125	1.2 (0.9–1.6)	119	1.7 (1.3–2.3)	1.7 (1.3–2.3)	4	8	1.3 (0.3–5.0)	7	1.1 (0.3–4.1)	0.8 (0.3–2.7)	0.9 (0.4–2.3)
**Milk yogurt (e.g., probiotic drinks)**													
less than once per week	4822	3699	1.0	1662	1.0	1.0	99	110	1.0	70	1.0	1.0	1.0
1–3 times per week	7912	7219	1.1 (1.0–1.2)	3955	1.1 (1.0–1.2)	1.1 (1.0–1.2)	193	227	0.9 (0.6–1.3)	223	1.4 (0.9–2.0)	1.5 (1.1–2.2)	1.4 (1.1–1.9)
4+ times per week	932	849	1.1 (0.9–1.2)	515	1.1 (0.9–1.3)	1.1 (0.9–1.3)	42	37	0.7 (0.4–1.2)	45	1.3 (0.7–2.2)	1.9 (1.1–3.4)	2.1 (1.4–3.3)
**Hand-shaken drinks (e.g., bubble milk tea)**													
less than once per week	9992	7769	1.0	3501	1.0	1.0	244	235	1.0	208	1.0	1.0	1.0
1–3 times per week	3507	3799	1.2 (1.1–1.2)	2457	1.4 (1.3–1.5)	1.4 (1.3–1.5)	85	126	1.3 (0.9–1.9)	118	1.2 (0.7–1.7)	0.9 (0.6–1.3)	0.6 (0.4–0.7)
4+ times per week	167	199	1.2 (0.9–1.5)	174	1.8 (1.4–2.2)	1.8 (1.4–2.3)	5	13	2.4 (0.8–7.6)	12	2.3 (0.7–7.2)	1.0 (0.4–2.5)	0.6 (0.3–1.2)
**Snacks**													
**Sweet snacks (e.g., cake and candies)**													
less than once per week	1705	991	1.0	366	1.0	1.0	46	42	1.0	24	1.0	1.0	1.0
1–3 times per week	8317	7249	1.4 (1.3–1.5)	3654	1.6 (1.4–1.8)	1.6 (1.4–1.9)	197	219	1.0 (0.6–1.7)	209	1.5 (0.8–2.6)	1.4 (0.8–2.5)	0.8 (0.5–1.2)
4+ times per week	3520	3432	1.6 (1.4–1.7)	2065	2.0 (1.8–2.3)	2.0 (1.8–2.3)	89	109	1.0 (0.6–1.8)	104	1.5 (0.7–2.9)	1.4 (0.8–2.7)	0.7 (0.4–1.3)
**Salty snacks (e.g., potato chips)**													
less than once per week	6190	4545	1.0	1977	1.0	1.0	150	146	1.0	104	1.0	1.0	1.0
1–3 times per week	6688	6427	1.1 (1.0–1.1)	3663	1.2 (1.1–1.3)	1.2 (1.1–1.3)	163	201	1.0 (0.7–1.5)	210	1.4 (0.9–2.0)	1.4 (0.9–2.0)	0.9 (0.7–1.2)
4+ times per week	704	720	1.1 (0.9–1.2)	451	1.1 (0.9–1.3)	1.1 (0.9–1.3)	19	24	1.1 (0.5–2.3)	23	1.1 (0.5–2.4)	1.0 (0.5–2.1)	0.7 (0.4–1.2)

Abbreviations: SECC: severe early childhood caries; aOR: adjusted odds ratio; CI: confidence interval. ^1^ aORs were adjusted for child’s age and gender and parent’s educational level, as well as variables in the table. ^2^ aORs for comparing sugar-sweetened beverage and snack intake between SECC children with native or immigrant participants.

**Table 4 ijerph-16-01047-t004:** Adjusted odds ratios of dental caries status associated with dental caries preventive behaviors in native and immigrant children.

Factor	Native						Immigrant						
	Caries-Free	Non-SECC		SECC		SECC vs. Non-SECC	Caries-Free	Non-SECC		SECC		SECC vs. Non-SECC	
	No.	No.	aOR ^1^ (95% CIs)	No.	aOR ^1^ (95% CIs)	aOR Ratio ^1^ (95% CIs)	No.	No.	aOR ^1^ (95% CIs)	No.	aOR ^1^ (95% CIs)	aOR Ratio ^1^ (95% CIs)	aOR ^1,2^ (95% CIs)
**Does your child need to be reminded to brush their teeth?**													
No	6876	5804	1.0	2747	1.0	1.0	167	183	1.0	172	1.0	1.0	1.0
Yes	6576	5803	1.2 (1.1–1.2)	3307	1.4 (1.3–1.5)	1.4 (1.3–1.5)	162	185	1.3 (0.9–1.8)	162	1.2 (0.8–1.7)	0.9 (0.6–1.3)	0.8 (0.6–1.1)
**Do you assist your child in brushing his/her teeth before sleep at night?**													
Yes	7256	5523	1.0	2722	1.0	1.0	161	136	1.0	99	1.0	1.0	1.0
No	6127	6012	1.1 (1.0–1.2)	3267	1.2 (1.1–1.3)	1.2 (1.1–1.3)	161	222	1.3 (0.9–1.8)	216	1.6 (1.1–2.3)	1.3 (0.9–1.8)	1.2 (0.9–1.6)
**Do you assist your child in cleaning his/her teeth by using dental floss?**													
Yes	5845	5072	1.0	2428	1.0	1.0	109	111	1.0	74	1.0	1.0	1.0
No	7406	6329	1.0 (0.9–1.1)	3513	1.1 (1.0–1.2)	1.1 (1.0–1.2)	222	250	1.1 (0.8–1.6)	251	1.4 (0.9–2.3)	1.3 (0.9–1.8)	1.6 (1.2–2.2)
**Do you take your child for semiannual dental visits?**													
No	5941	4774	1.0	2506	1.0	1.0	187	199	1.0	197	1.0	1.0	1.0
Yes	7626	6911	1.2 (1.1–1.3)	3565	1.5 (1.4–1.6)	1.5 (1.4–1.6)	145	174	1.5 (0.9–2.1)	140	1.7 (1.1–2.5)	1.1 (0.7–1.7)	1.2 (0.9–1.6)
**Does your child receive fluoride varnish application every 6 months?**													
Yes	8685	7330	1.0	3626	1.0	1.0	186	187	1.0	145	1.0	1.0	1.0
No	4875	4343	1.1 (1.0–1.2)	2444	1.3 (1.2–1.5)	1.3 (1.2–1.5)	145	179	1.2 (0.8–1.7)	188	1.4 (0.9–2.1)	1.2 (0.9–1.8)	1.1 (0.8–1.5)

Abbreviations: SECC: severe early childhood caries; aOR: adjusted odds ratio; CI: confidence interval. ^1^ aORs were adjusted for child’s age and gender and parent’s educational level, as well as the variables in the table. ^2^ aORs for comparing dental caries preventive behaviors between the native and immigrant groups among SECC children.

**Table 5 ijerph-16-01047-t005:** Combined effects of dental caries preventive behaviors and sugar-sweetened beverage or snack intake on the dental caries status in native participants.

Factor		Caries-Free	Non-SECC ^1^		SECC ^1^	
		No.	No.	aOR ^2^ (95% CIs)	No.	aOR ^2^ (95% CIs)
**Parental assistance in brushing teeth**	**SSB intake, time/week**					
Yes	less than once	2207	1372	1.0	532	1.0
Yes	1–3 times	5012	4105	1.3 (1.2–1.4)	2160	1.7 (1.5–1.9)
Yes	4+ times	37	46	2.1 (1.3–3.3)	30	3.0 (1.8–5.0)
No	less than once	1594	1306	1.1 (1.0–1.3)	573	1.3 (1.1–1.3)
No	1–3 times	4479	4636	1.4 (1.3–1.6)	2619	2.0 (1.8–2.2)
No	4+ times	54	70	1.9 (1.3–2.7)	75	4.8 (3.3–7.1)
**Fluoride varnish application**	**SSB intake, time/week**					
Yes	less than once	2497	1735	1.0	711	1.0
Yes	1–3 times	6144	5532	1.3 (1.2–1.4)	2866	1.6 (1.4–1.7)
Yes	4+ times	44	63	2.2 (1.5–3.3)	49	3.8 (2.4–5.8)
No	less than once	1351	975	1.0 (0.9–1.1)	402	1.0 (0.9–1.1)
No	1–3 times	3471	3316	1.3 (1.2–1.4)	1985	1.8 (1.6–2.0)
No	4+ times	53	52	1.3 (0.8–2.0)	57	3.1 (2.1–4.7)
**Use of dental floss**	**Snack intake, time/week**					
Yes	less than once	632	372	1.0	95	1.0
Yes	1–3 times	4042	3466	1.4 (1.3–1.7)	1630	2.5 (2.0–3.2)
Yes	4+ times	1094	1174	1.9 (1.6–2.2)	666	3.9 (3.0–4.9)
No	less than once	634	363	1.0 (0.8–1.2)	138	1.4 (1.0–1.8)
No	1–3 times	5018	4267	1.4 (1.2–1.6)	2244	2.7 (2.1–3.3)
No	4+ times	1634	1596	1.7 (1.5–2.0)	1086	4.1 (3.2–5.1)
**Fluoride varnish application**	**Snack intake, time/week**					
Yes	less than once	851	485	1.0	145	1.0
Yes	1–3 times	5972	5026	1.4 (1.3–1.7)	2469	2.3 (1.9–2.8)
Yes	4+ times	1730	1718	1.8 (1.6–2.1)	964	3.2 (2.6–3.9)
No	less than once	436	265	1.1 (0.9–1.3)	92	1.1 (0.8–1.5)
No	1–3 times	3301	2883	1.5 (1.3–1.7)	1496	2.3 (1.9–2.8)
No	4+ times	1067	1130	1.9 (1.6–2.2)	821	4.0 (3.3–4.9)

Abbreviations: SECC: severe early childhood caries; aOR: adjusted odds ratio; CI: confidence interval; SSBs: sugar sweetened beverages. ^1^ Non-SECC group is defined as 0 < dmft < 4 for an age of 3 years, 0 < dmft < 5 for an age of 4 years, and 0 < dmft < 6 for an age of 5 years. SECC group is defined as dmft ≥4 for an age of 3 years, ≥5 for an age of 4 years, and ≥6 for an age of 5 years. ^2^ aORs were adjusted for child’s age and gender and parent’s educational level.

**Table 6 ijerph-16-01047-t006:** Combined effects of dental caries preventive behaviors and sugar-sweetened beverage or snack intake on the dental caries status in immigrant participants.

Factor		Caries-Free	Non-SECC ^1^		SECC ^1^	
		No.	No.	aOR ^2^ (95% CIs)	No.	aOR ^2^ (95% CIs)
**Parental assistance in brushing teeth**	**SSB intake, time/week**					
Yes	less than once	45	28	1.0	16	1.0
Yes	1–3 times	114	108	1.5 (0.8–2.6)	82	1.8 (0.9–3.6)
Yes	4+ times	2	0	–	1	1.8 (0.2–21.6)
No	less than once	40	39	1.2 (0.6–2.5)	32	1.6 (0.7–3.4)
No	1–3 times	119	178	2.0 (1.1–3.4)	180	3.2 (1.7–6.1)
No	4+ times	2	5	3.0 (0.5–17.2)	4	3.3 (0.5–20.3)
**Parental assistance in brushing teeth**	**Snack intake, time/week**					
Yes	less than once	16	16	1.0	2	1.0
Yes	1–3 times	106	83	0.6 (0.3–1.4)	68	3.7 (0.8–17.3)
Yes	4+ times	38	33	0.7 (0.3–1.8)	29	4.7 (0.9–22.7)
No	less than once	17	14	0.6 (0.2–1.6)	15	4.1 (0.8–21.7)
No	1–3 times	104	147	1.0 (0.4–2.1)	136	6.2 (1.3–28.4)
No	4+ times	38	59	1.0 (0.4–2.4)	63	8.2 (1.7–38.8)

Abbreviations: SECC: severe early childhood caries; aOR: adjusted odds ratio; CI: confidence interval; SSBs: sugar sweetened beverages. ^1^ Non-SECC group is defined as 0 < dmft < 4 for an age of 3 years, 0 < dmft < 5 for an age of 4 years, and 0 < dmft < 6 for an age of 5 years. SECC group is defined as dmft ≥4 for an age of 3 years, ≥5 for an age of 4 years, and ≥6 for an age of 5 years. ^2^ aORs were adjusted for child’s age and gender and parent’s educational level.

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
