# Peer review of "Immigrant–Native Differences in Sugar-Sweetened Beverage and Snack Consumption and Preventive Behaviors Associated with Severe Early Childhood Caries: A Large-Scale Survey in Taiwan"

_ijerph, 2019, doi:10.3390/ijerph16061047_

Round 1
Reviewer 1 Report
This was an informative paper with adequate charts that help substantiate the discussion and conclusion. I was impressed by the thoroughness of the information provided that gets to the heart of the problem of early childhood decay. The impact of poverty and displacement of populations that include children is important globally. The convenience stores that sell sugary snacks and beverages is an issue globally as well. I did not see a reference to the addictive associations of sugar as another variable which should be stressed in the preventive education for parents. You did point out that the issue of sugar consumption increases with age. At this junction, the children begin to make their own decisions about food consumption. We find this to be the general rule. Parental education, cultural sensitivity and understanding the parent's oral issues are significant when preparing for the next important steps toward prevention.
References 28 and 31, do they specifically point to flossing for toddlers? Are there alternatives to flossing at this young age, or is it really necessary?
What are the barriers for immigrant mothers taking their children to the dentist for checkup and Fl varnish? Is it the tradition or are there other reasons that you only mention "mothers" taking children to the doctor. What role do the fathers play; or what percentage of the parents are together?
Does the Taiwan government provide the same services to the immigrants as they do for the natives? What are these variations? What is the difference in their standard of living?
Author Response
Response to Reviewer 1 Comments
Point 1: This was an informative paper with adequate charts that help substantiate the discussion and conclusion. I was impressed by the thoroughness of the information provided that gets to the heart of the problem of early childhood decay. The impact of poverty and displacement of populations that include children is important globally. The convenience stores that sell sugary snacks and beverages is an issue globally as well. I did not see a reference to the addictive associations of sugar as another variable which should be stressed in the preventive education for parents. You did point out that the issue of sugar consumption increases with age. At this junction, the children begin to make their own decisions about food consumption. We find this to be the general rule. Parental education, cultural sensitivity and understanding the parent's oral issues are significant when preparing for the next important steps toward prevention.
Response 1: Thanks for the suggestion. We have been citation the reference about sugar addictive, also discussed children chose sugary food, the parent with low education, and less information of dental caries prevention, these risk factors cause children’s dental caries. (lines 253-261 highlight in red)
Point 2: References 28 and 31, do they specifically point to flossing for toddlers? Are there alternatives to flossing at this young age, or is it really necessary
Response 2: References 28 and 31 reported that behaviors for preventing SECC include parental assistance in teeth brushing at night and the use of dental floss. Flossing in toddlers is valuable for the prevention of interdental caries. For young children, suggested parents use dental floss or dental floss stick to clean interdental surfaces.
Point 3: What are the barriers for immigrant mothers taking their children to the dentist for checkup and Fl varnish? Is it the tradition or are there other reasons that you only mention "mothers" taking children to the doctor. What role do the fathers play; or what percentage of the parents are together?
Response 3: Women from Southeast Asian countries have been migrating to Taiwan since 1987. This particular group of immigrant women is highly susceptible and vulnerable to health problems because of language barriers, cultural conflicts, social and interpersonal isolation, and lack of support systems. We mention “immigrant mother” in the main text. In this study, the questions of preventive behaviors were asked “parents assistance in...”; because we believe that no matter the mother or father, they are important for a child’s health promotion.
Point 4: Does the Taiwan government provide the same services to the immigrants as they do for the natives? What are these variations? What is the difference in their standard of living?
Response 4: The National Health Insurance actively promotes free fluoride varnish every 6 months for children aged less than 6 years, the dentists have been providing this preventive behavior in kindergarteners. Children from minority families who might not go to a kindergartener and the health information are not readily available for people in low socioeconomic segments, particularly for immigrant mothers with a language barrier. Therefore, children born to a foreign spouse who uses oral healthcare services less frequently.

Reviewer 2 Report
Interesting reading but could benefit from some English editing and considering a few suggestions:
Introduction.
Needs re-structuring to make clear to the readers what the main purpose of the study is. The authors describe with greater detail well-known findings from previous studies (i.e. risk behaviours for dental caries) instead of focusing on the main variables for their study and the rationale for them. They even jump into a conclusion ('cross-cultural caries prevention programs for immigrant and native young children should focus on children at a high risk of dental caries' sic) which, if already known, could mean their study is no longer needed.
Methods
Please include details on sampling procedures and sample size calculation.
How was the comsumption of snacks and sweetened drinks recorded? Were the parents asked to remember what happened the week before or a diet diary was used? Please provide details and also the reasoning (or reference) for the weekly (instead of daily) cut-off points.
Try to clarify which variables were considered exposure, outcome and which were confounders.
Results
When reading this section it seems the questions were different to those explained in the methods section, please clarify which one is the right one and modify accordingly.
The last paragraph of this section is a bit confusing: did the authors find any significant interaction between the variables of interest?
The tables need to be re-structured; perhaps 3 would be enough to present the results:
Description of the sample in terms of demographic and SES characteristics: for caries-free children (presenting data for native and immigrant groups and the p value when comparing these two), non SECC (as above) and SESS (as above)
Description of the sample in terms of behaviours organised as above
Regression models (if any significant interactions were found consider another table for stratified analysis).
Discussion
Will need to be completely re-written based on above comments.
Please do not forget to include the limitations of the study.
Author Response
Response to Reviewer 2 Comments
Interesting reading but could benefit from some English editing and considering a few suggestions:
Response: This manuscript has undergone English language editing by Multidisciplinary Digital Publishing Institute.
Introduction
Point 1: Needs re-structuring to make clear to the readers what the main purpose of the study is. The authors describe with greater detail well-known findings from previous studies (i.e. risk behaviours for dental caries) instead of focusing on the main variables for their study and the rationale for them. They even jump into a conclusion ('cross-cultural caries prevention programs for immigrant and native young children should focus on children at a high risk of dental caries' sic) which, if already known, could mean their study is no longer needed.
Response 1: We have revised the section. (lines 68-80 highlight in red)
Methods
Point 2: Please include details on sampling procedures and sample size calculation.
Response 2: This large-scale survey of oral health conditions. All children aged 3-5 years who has live in Taipei were recruited from the 52 community health centers and from 659 preschools (149 public, 510 private) in 12 districts of Taipei city. After excluded the invalid questionnaires, the final datasets, obtained from 32,611 child-parent pairs. We have revised the recruitment of participants. (lines 84-92 highlight in red)
Point 3: How was the comsumption of snacks and sweetened drinks recorded
Response 3: Children’s SSBs and snacks consumption were measured based on their parents’ reports. The number of times in the previous week that the children had consumed the following items was reported: (1) soft drinks (e.g., soda, coke); (2) milk yogurt (e.g., probiotic drinks); (3) hand-shaken drinks (e.g., bubble milk tea); (4) sweet snacks (e.g., cake, candies); and (5) salty snacks (e.g., potato chips).The question was worded as follows: “How many times did your child consume in the previous week?” (with possible answers being “less than once,” “one to three times”, “four to six times,” and “seven times or more “). (lines 116-126 highlight in red)
Point 4: Were the parents asked to remember what happened the week before or a diet diary was used?
Response 4: We did not ask parents to use their child’s diet diary. Parents recall their child’s consumption frequencies of SSBs and snacks last week. We described this limitation in the discussion. (lines 295-300 highlight in red)
Point 5: Please provide details and also the reasoning (or reference) for the weekly (instead of daily) cut-off points.
Response 5: We refer to van Ansem et al. to measure children’s consumption frequencies of SSBs and snacks. (lines 117-118 highlight in red)
References
15. van Ansem, W.J.; van Lenthe, F.J.; Schrijvers, C.T.; Rodenburg, G.; van de Mheen, D. Socio-economic inequalities in children's snack consumption and sugar-sweetened beverage consumption: the contribution of home environmental factors. British Journal of Nutrition 2014, 112, 467-476.
Point 6: Try to clarify which variables were considered exposure, outcome and which were confounders.
Response 6: We have revised the “materials and Methods” section; and added sub-section titled “2.3. Dependent variable” (lines 106-114) and “2.4. Independent variables” (lines 115-126), besides we describe confounders in “statistical analysis” (line 154 highlight in red).
Results
Point 7: When reading this section it seems the questions were different to those explained in the methods section, please clarify which one is the right one and modify accordingly.
Response 7: We have revised the terms of independent variables in “Results” and tables.
Point 8: The last paragraph of this section is a bit confusing: did the authors find any significant interaction between the variables of interest?
Response 8: The last paragraph explained the significant combined effects of caries preventive behaviors and SSBs or snacks intake on the dental caries status in children. But these variables did not show the interaction effects.
Point 9: The tables need to be re-structured; perhaps 3 would be enough to present the results:
Response 9: After a discussion by the authors, we’d like to keep these six tables, every table has different research finding for readers. Following reviewer's suggestions, we have described the terms of the variable more clearly, as same as in "Independent variables".
Point 10: Description of the sample in terms of demographic and SES characteristics: for caries-free children (presenting data for native and immigrant groups and the p value when comparing these two), non SECC (as above) and SESS (as above)
Response 10: We analyzed the demographic variables between native and immigrant groups among children caries-free, non-SECC, and SECC. (Table 2; lines 170-172 highlight in red)
Point 11: Description of the sample in terms of behaviours organised as above
Response 11: We have revised the terms of behaviors in Table 4, as the description of “caries preventive behaviors” section in the main text. (Table 4, lines 131-135 highlight in red)
Point 12: Regression models (if any significant interactions were found consider another table for stratified analysis).
Response 12: There was no interaction effect of caries preventive behaviors and SSBs or snacks intake associated with SECC among the two groups.
Discussion
Point 13: Will need to be completely re-written based on above comments.
Response 13: The Discussion section has been re-written and removed some information without relevance.
Point 14: Please do not forget to include the limitations of the study.
Response 14: We mentioned the study limitations in the last paragraph of the Discussion section. (lines 295-308 highlight in red)
Reviewer 3 Report
It is clear that the topic of this study is of interest but it needs a revision. This manuscript is relatively interesting to read but rather difficult to follow. There were a lot of information, mainly in “Results” section that should be shortened. There some points that must be cleared out.
Specific comments
Introduction:
This section should be written more clearly to justify the reason of a large scale survey, as well as the importance and relationship between caries preventive behaviour and consumption frequency of SSBs of immigrant or native children. Clarify the aim of the study and the null hypothesis. There are paragraphs repeated in “Discussion”, it should be avoided.
Materials and methods:
-How the sample was determined?
-How many dentists did the examination? Kappa values, intra and inter-examiners reliability?
-Clarify how was the methodology of SSBs and snacks consumption record.
Results:
-Remove table 3. The association between SSBs and snacks consumptions and dental caries. Do not bring an additional information. It is evident that regardless of children (native or immigrant) the more frequently consumptions, the more risk of caries development. The point is if the caries preventive behaviuor modify that risk in a certain condition.
Discussion:
This section should be rewritten. There some information without relevance. As mentioned above, there are paragraphs repeated in introduction.
Author Response
Response to Reviewer 3 Comments
It is clear that the topic of this study is of interest but it needs a revision. This manuscript is relatively interesting to read but rather difficult to follow. There were a lot of information, mainly in “Results” section that should be shortened. There some points that must be cleared out.
Response: Thanks. We do our best to short the "results" section and keep the important findings.
Specific comments
Introduction:
Point 1: This section should be written more clearly to justify the reason of a large scale survey, as well as the importance and relationship between caries preventive behaviour and consumption frequency of SSBs of immigrant or native children. Clarify the aim of the study and the null hypothesis.
Response 1: Taipei is the capital city of Taiwan, also the target city of health policy implementation, and providing accessibility and availability of dental medical resources. A large-scale survey in Taipei city, that could be excluded due to the confounder shortage of medical resources. This study recruited a large representative group of native children and children born to a foreign spouse. The study aims to investigate immigrant–native differences in relation to SSBs and snacks intake and caries preventive behaviors, associated with SECC in Taipei, to develop dental caries prevention and intervention strategies for young children.
(lines 52-56, 68-80 highlight in red)
Point 2: There are paragraphs repeated in “Discussion”, it should be avoided.
Response 2: We have revised the section and remove a similar description.
Materials and methods
Point 3: How the sample was determined?
Response 3: These children aged 3-5 years were recruited from the 52 community health centers and from 659 preschools (149 public, 510 private) in 12 districts of Taipei city. After excluding the invalid questionnaires, the final datasets, obtained from 32,611 child-parent pairs. We have revised the recruitment of participants. (lines 84-92 highlight in red)
We added the “data collection” section. (lines 137-144 highlight in red)
Point 4: How many dentists did the examination? Kappa values, intra and inter-examiners reliability?
Response 4: The dentists did not examine the intra and inter reliability before performing the dental examination on the children, but they were well-trained in pediatric dentistry. Therefore, the dental caries diagnosis was reliable. We have added this study limitation in the discussion. (lines 300-303 highlight in red)
Point 5: Clarify how was the methodology of SSBs and snacks consumption record.
Response 5: Children’s SSBs and snacks consumption were measured based on their parents’ reports. The number of times in the previous week that the children had consumed the following items was reported: (1) soft drinks (e.g., soda, coke); (2) milk yogurt (e.g., probiotic drinks); (3) hand-shaken drinks (e.g., bubble milk tea); (4) sweet snacks (e.g., cake, candies); and (5) salty snacks (e.g., potato chips).The question was worded as follows: “How many times did your child consume in the previous week?” (with possible answers being “less than once,” “one to three times”, “four to six times,” and “seven times or more “). (lines 116-126 highlight in red)
Results
Point 6: Remove table 3. The association between SSBs and snacks consumptions and dental caries. Do not bring an additional information. It is evident that regardless of children (native or immigrant) the more frequently consumptions, the more risk of caries development. The point is if the caries preventive behaviuor modify that risk in a certain condition.
Response 6: Table 3 in this study presents the children’s caries status associated with SSBs and snacks consumption in native and immigrant groups. It is important to descript the combined effects has shown in Table 5 and Table 6. Another reviewer thinks that Table 3 would enough to present the study results. We sincerely hope to keep Table 3.
Discussion
Point 7: This section should be rewritten. There some information without relevance. As mentioned above, there are paragraphs repeated in introduction.
Response 7: Thanks for suggestions. We have revised the section.
Round 2
Reviewer 2 Report
Thank you for your hard work addressing the comments received. However, it is mentioned that the aim of your study is to compare native and immigrant population in terms of behaviours and ECC; the tables presented are, from my point of view, failing to do so as the results for each group are presented separately and there is no statistical comparison between them. Perhaps if you include the variable 'immigration status (born to native or immigrant parent)' in the regression models you could achieve the expressed aim.
Author Response
Point 1: Thank you for your hard work addressing the comments received. However, it is mentioned that the aim of your study is to compare native and immigrant population in terms of behaviours and SECC; the tables presented are, from my point of view, failing to do so as the results for each group are presented separately and there is no statistical comparison between them. Perhaps if you include the variable 'immigration status (born to native or immigrant parent)' in the regression models you could achieve the expressed aim.
Response 1: Thanks for the suggestion. We had compared the SSBs and snacks intake and dental caries preventive behaviors between the immigrant and native participants. The results showed on the Table 3 and Table 4. In the main text, we had revised with highlight in red.
Reviewer 3 Report
Thank you for the author response to the comments. The reviewed version of this manuscript has been improved significantly.
Author Response
Point 1: Thank you for the author response to the comments. The reviewed version of this manuscript has been improved significantly.
Response 1: Thank you.

Round 3
Reviewer 2 Report
Good work at addressing reviewer's comments